# What combination of interventions can optimise HIV prevention for adolescent girls and young women? Cohort analysis of DREAMS participation in urban and rural Kenya

Faith Magut[1*], Sarah Mulwa[1], Annabelle Gourlay[1], Vivienne Kamire[2], Jane Osindo[3], Elvis O. A. Wambiya[4], Moses Otieno[2], Elona Toska[5], Jane Ferguson[6], Brendan Maughan-Brown[7], Daniel Kwaro[2], Abdhalah Ziraba[3], Isolde Birdthistle[1], Sian Floyd[1]

1 Faculty of Epidemiology of Population Health, London School of Hygiene and Tropical Medicine, London, United Kingdom, 2 Center for Global Health Research, Kenya Medical Research Institute, Kisumu, Kenya, 3 Health and Systems for Health, African Population and Health Research Center, Nairobi, Kenya, 4 School of Health and Related Research, University of Sheffield, Sheffield, United Kingdom, 5 Centre for Social Science Research, University of Cape Town, Cape Town, South Africa, 6 Independent Consultant, Geneva, Switzerland, 7 Southern Africa Labour and Development Research Unit, University of Cape Town, Cape Town, South Africa

* Faith.Magut@lshtm.ac.uk

## Abstract

Comprehensive intervention packages are recommended to address multiple sources of HIV risk for adolescent girls and young women (AGYW). DREAMS is a multi-component HIV prevention program designed to reduce HIV incidence among AGYW. We conducted a prospective cohort study among AGYW aged 13–22 years, randomly selected in rural Gem and urban Nairobi informal settlements followed from 2017/2018–2019. AGYW were classified into three groups: (1) invited to DREAMS and received a "complete" package, (2) invited and received a "partial" package, or (3) not invited to DREAMS. We defined the "complete" package as 4–5 primary interventions in Gem and 5 in Nairobi: the "partial" package as 3 specific interventions in Gem and any 3–4 interventions in Nairobi. We used propensity score-adjusted logistic regression to estimate the causal effect of DREAMS on outcomes under three counterfactual scenarios: all AGYW accessed the complete package, all accessed a partial package, or none were invited. In Nairobi, 1081 AGYW were enrolled. By 2019, 26% accessed the complete package and 32% accessed the partial package. Among those receiving the complete package, there was increase in HIV status knowledge(24.8% [95%CI:16.4,32.6]),social support(13.9% [95%CI:3.3,23.6]) and self-efficacy(10.3% [95%CI:0.5,20.4]) and a decrease in the proportion with ≥2 lifetime partners(-8.0% [95%CI:-15.9,0.0]). In Gem, 1171 AGYW were enrolled. By 2019, 24% received the complete package and 21% received the partial package. We found evidence of an increase in HIV status knowledge(10.0% [95%CI:4.5,15.2]),

**Data availability statement:** Data underlying published results will be accessible and open, subject to a transition period (available from the London School of Hygiene and Tropical Medicine data repository https://datacompass. lshtm.ac.uk by contacting researchdatamanage-ment@lshtm.ac.uk), as per the Open Access Policy of the Bill & Melinda Gates Foundation.

**Funding:** IB and SF were funded for the impact evaluation of DREAMS program by the Bill and Melinda Gates Foundation (OPP1136774, http://www.gatesfoundation.org). The NUHDSS in which the DREAMS Impact Evaluation is nested is partly funded by SIDA: Grant #54100113. MS was funded through the National Institutes of Health under award number 5R01MH114560–03 and Africa Health Research Institute grant from the Wellcome Trust (082384/Z/07/Z). Funders had no role in the design, implementation, evaluation, or writing of the manuscript.

**Competing interests:** The authors have declared that no competing interests exist.

social support(27.2% [95%CI:19.2,35.5]) and a decrease in condomless sex(-9.1% [95%CI:-13.6,-4.1]), and the proportion with ≥2 lifetime partners(-7.6% [95%CI:-12.4,-2.2]) for the complete package. Among those receiving the partial package, there was a decrease in condomless sex(-12.2% [95%CI: -17.0,-6.4]), and an increase in self-efficacy(8.0% [95%CI:0.0,17]). A package of 4–5 primary DREAMS interventions had positive impacts on multiple HIV-related outcomes in both settings. A partial package was effective in Gem, but not in Nairobi, suggesting the need for context-specific intervention strategies.

## Introduction

The Global AIDS strategy 2021–2026 recommends intensifying HIV prevention for the populations and places experiencing the greatest need in order to end AIDS as a public health threat by 2030. This includes adolescent girls and young women (AGYW) aged 15–24 years [1]. In most settings with high HIV prevalence and inci-dence, AGYW are considered a 'priority population', that is, a group within the gen-eral population that is in need of HIV prevention due to disproportionate risk [2]. In Eastern and Southern Africa, for example, AGYW on average experience three times the risk of HIV infection compared to their male peers, and are disproportionately affected by the epidemic in this region [1,3].

The Joint United Nations Programme on HIV/AIDS (UNAIDS) advocates the scale-up of combination prevention programmes to reach 95% of key and priority populations. One of the five central 'pillars of prevention' focuses on AGYW, recom-mending a combination prevention package that integrates interventions across 3 dimensions. These 3 dimensions are: (1) behavioural interventions including compre-hensive sexuality education, both in and out of school; (2) biomedical interventions including HIV and sexual and reproductive health services, and antiretroviral-based prevention such as Pre-Exposure Prophylaxis (PrEP); and (3) structural interventions to modify harmful gender norms, end gender-based discrimination, mitigate inequali-ties and violence, improve social protection and support economic empowerment [4].

In the past decade, significant investments have been made in HIV prevention for AGYW, with U.S. President's Emergency Plan for AIDS Relief (PEPFAR) and Global Fund resources dedicated to delivering packages of evidence-based interventions, covering each of the three dimensions. PEPFAR's large investment in the DREAMS Partnership (for Determined, Resilient, Empowered, AIDS-free, Mentored and Safe lives) focuses on the UNAIDS second pillar of HIV prevention among AGYW by promoting an extensive 'core package' of interventions across more than 150 dis-tricts in 15 countries. The 'core package' includes interventions selected to empower AGYW and reduce their HIV risk, as well as contextual programmes to strengthen the families of AGYW, mobilize communities for norms change, and lower the HIV risk for men in the age range that includes most of the sex partners of AGYW [5].

In an independent evaluation of DREAMS in Kenya, we enrolled and followed representative cohorts of AGYW in urban (Nairobi) and rural (Gem) settings between

2017/18 and 2019. We found high programme reach, with most AGYW invited into DREAMS by 2019 [6]. Since DREAMS was intended as a coherent and integrated package, comprising multiple interventions for which there was already evidence of their individual effectiveness, we set out *a priori* to compare outcomes among those who were and were not invited into DREAMS during 2016–2018. This enabled us to assess the overall impact of the package, acknowledging that there was variation among DREAMS invitees in how many interventions they actually received. We found that over 70% of invitees had received ≥3 interventions by 2019 [6]. DREAMS led to increased knowledge of HIV status and social support in both settings, and it increased self-efficacy among younger AGYW (age 13–17 years) in the rural setting. Additionally, DREAMS contributed to a reduction in condomless sex and in the number of sexual partners among younger AGYW in the rural setting; and a reduction in condomless sex among sexually active older AGYW (age 18–22 years) in the urban setting [7,8].

Questions around optimal intervention packages remain critical in HIV prevention programming, given constrained resources [9]. Evidence of the effect of combinations of DREAMS interventions on various outcomes has been demonstrated in other settings, indicating that benefits can be gained from streamlined packages of interventions. An evaluation of DREAMS in South Africa found that accessing 3 or more DREAMS interventions (out of 10) was associated with increased HIV testing, attaining higher HIV knowledge index scores, and access to contraceptives, compared to those who did not access DREAMS interventions [10]. An evaluation of DREAMS in Lesotho found that AGYW who received 2 or more interventions reported lower levels of sexual risk and higher levels of self-efficacy compared to matched peers who did not receive any interventions [11]. Another study in Kisumu, Kenya, found that exposure to a combination of 2 or more DREAMS interventions, compared to none, can reduce sexual risk behaviour outcomes among AGYW. In this study, a combination of schooling support, PrEP awareness, HIV education, and gender-based violence prevention increased consistent condom use among AGYW, and exposure to the youth fund program, violence prevention program, schooling support, and parenting programming reduced transactional sex [12]. To add to this evidence base, we analysed the impact of partial or more complete packages of DREAMS interventions on a range of HIV prevention and psycho-social outcomes among AGYW in urban and rural settings in Kenya, using data from the DREAMS impact evaluation project [13].

## Methods

### Research settings

The impact evaluation study was conducted in two settings in Kenya: Nairobi (urban setting) and Gem in western Kenya (rural setting). In Nairobi, recruitment was conducted in two urban informal settlements, situated within the Nairobi Urban and Health and Demographic Surveillance System (NUHDSS) which was established in 2002 and it covers approximately 90,000 individuals and 33,000 households [14]. Gem sub-county in western Kenya is included within the Kenya Medical Research Institute (KEMRI) and Center for Disease Control and Prevention (CDC) Health and Demographic Surveillance System (HDSS) that was established in 2002, covering approximately 260,000 residents and 55,000 households [15].

### Study design

The study design was a prospective observational closed cohort, with a random sample of AGYW selected from a population-wide sampling frame drawn from the HDSS in each setting. Stratified sampling was used with AGYW divided into two age groups: 15–17 and 20–22 years in Nairobi, and 13–17 and 18–22 years in Gem, with a target sample of 500 AGYW in each age group. Further details can be found in the study protocol [13].

AGYW were enrolled into the cohort in Nairobi in 2017, with follow-up in both 2018 and 2019. In Gem, enrolment was in 2018, with follow-up in 2019. At both enrolment and follow-up, questionnaires were used to collect data on various household and individual characteristics, knowledge of HIV status, sexual behaviour (including condom use and number of sexual partners), social support, generalised self-efficacy, and participation in the DREAMS interventions. The questionnaire

administration was preceded by a consenting and assenting process. All interviews were conducted face-to-face in the AGYW households by trained research assistants. To ensure privacy and confidentiality, interviews were held in a private space within the home, away from other household members whenever possible.

## DREAMS implementation context and DREAMS interventions

Kenya was identified as one of the priority countries for the implementation of DREAMS by the US President's Emergency Plan for AIDS Relief (PEPFAR). DREAMS was introduced to priority countries in 2016,with implementing partners contracted by the U.S. Government to deliver the core package to benefit vulnerable girls and young women aged 10–24 years [16]. The core package of DREAMS interventions comprises a range of initiatives, from HIV testing, condom promotion, and provision of pre-exposure prophylaxis (PrEP) to social asset building, parenting programmes, educational subsidies, economic empowerment initiatives, and community mobilisation efforts. Social asset building aims to strengthen social capital through mentor-led meetings held regularly in "safe spaces," which are typically private, girls-only spaces such as community centers, churches, or schools. In these spaces, AGYW can access support, engage in curriculum-based programmes, and be linked to other services. In 2017, national prioritisation exercises led to the designation of 'primary' and 'secondary' interventions for AGYW (Fig 1). Primary interventions constituted a minimum set of interventions that all AGYW in their age-group [10–24] should receive, if invited to participate in DREAMS [5,13]. There

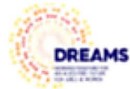

**KENYA DREAMS LAYERING TABLE**

| | AGE GROUPS (years) | 10-14 | 15-19 | 20-24 | Notes |
|---|---|---|---|---|---|
| **INDIVIDUAL** | **Primary Individual Interventions** | • Social Asset Building<br>• School-based HIV & violence prevention<br>• Financial capability training | • Social Asset building<br>• HIV & violence prevention[1]<br>• Condom education and demonstration<br>• HTS<br>• PrEP IEC (18-19)<br>• Contraception IEC[1]<br>• Financial capability training | • Social Asset building<br>• HIV and violence prevention<br>• Condom education and demonstration<br>• HTS<br>• PrEP Education<br>• Contraception IEC[1]<br>• Financial capability training<br>• | [1]Offered in community setting only due to MOE guidelines.<br><br>PREP IEC offered to 18-24 year olds only |
| | **Secondary Individual Interventions** | • HTS/active linkage<br>• Post-violence care<br>• Education subsidies | • Condom provision<br>• PrEP Provision[2]<br>• Contraceptive Mix (Provision)[3]<br>• Post-violence Care<br>• Combination Socio-economic Approaches<br>• Education subsidies[1]<br>• Cash transfer | • Condom provision<br>• PrEP Provision[2]<br>• Contraceptive Mix (Provision)[3]<br>• Post-violence care<br>• Combined socio-economic approaches<br>• Cash transfer | [2] Only for AGYW ages 18-19 years<br><br>[3]Contraceptive Mix includes provision and active referrals) |
| | **Range Individual Level Interventions** | 3-6 | 7-14 | 7-13 | |
| | | **DREAMS SNUs** | | | |
| **CONTEXTUAL** | **Contextual Level Interventions** | • Parenting/Caregiver Programming<br>• Community mobilization & Norms Change<br>• Reducing risk of male sex partners (link to Condoms, HTS, VMMC, HIV Treatment) | | | |
| | **Total Contextual Level Interventions** | 3 | | | |

**Determined** **Resilient** **Empowered** **AIDS-Free** **Mentored** **Safe**

**Fig 1. Layering of DREAMS interventions in Kenya.**

were seven primary interventions for AGYW aged 15–24 years: (1) HIV testing services; (2) Social asset building; (3) School-based or community-based HIV and violence prevention; (4) Contraceptive mix (sexual reproductive health); (5) Condom promotion and/or education; (6) PrEP education; and (7) Financial capability training. Secondary interventions were to be offered based on AGYW's circumstances and needs, for example: post-violence care for victims of sexual violence; educational subsidies to support retention in school, promote school re-enrolment, and facilitate the transition to secondary education; or oral PrEP for those at highest sexual risk [13,17].

With this approach, AGYW aged 15–24 years would receive between 7–14 interventions during their participation in DREAMS. The invitation to participate in DREAMS was targeted and was based on the socio-economic and sexual risk vulnerabilities of AGYW, including factors such as pregnancy (current or ever), food insecurity, low household socioeconomic status, and being out of school or orphaned. These characteristics were used to classify AGYW into various risk profiles, and vulnerable AGYW were identified using the `Girl Roster' census method, referred to previously [16].

**Measures**

**Independent variable: exposure to DREAMS.** Our exposure of interest was a composite variable indicating whether an AGYW had been invited to participate in DREAMS interventions by 2018 (yes or no) and the number of primary interventions they accessed by 2019, based on self-reported responses. Uptake of interventions by 2019 was defined based on AGYW self-reports of whether they had accessed each intervention in the past 12 months, as captured in the 2017/2018 or 2019 survey rounds. This definition captured intervention exposure that occurred prior to outcome measurement at the 2019 endline. As described previously, there were seven primary interventions. However, for the purposes of this analysis, we grouped contraceptive mix, condom promotion and/or education, and PrEP education or provision into one category acknowledging that anyone who received at least one of these interventions would have received key education on sexual and reproductive health. This resulted in five primary interventions for our analysis: (1) HIV testing and counselling services (including facility-based, mobile clinics, home-based testing, testing in safe spaces, self-testing, testing with a sexual partner, and HIV testing through work/employment); (2) Social asset building; (3) School-based or community-based HIV and violence prevention; (4) Contraceptive mix (sexual reproductive health), condom promotion and/or education, and PrEP education and/or provision; and (5) Financial capability training. In addition, previous analysis indicated that the uptake of the primary interventions differed between Nairobi and Gem (for example, in Gem 2% of those aged 13–17 years at enrolment and 5% of those aged 18–22 years accessed all seven primary interventions; as against 12% of those aged 15–17 and 15% of those aged 18–22 in Nairobi) [6] therefore, the specific combinations used in this analysis also differed by setting (S1 Table).

Based on binary exposure to DREAMS and participation in the five primary interventions, we generated a composite exposure variable with four categories: (a) Not invited to DREAMS by 2018 (referred to as "non-invitee" henceforth); (b) Invited by 2018 and accessed 0, 1, or 2 primary interventions; (c) Invited and accessed a partial primary package (3 or 4 of any of the five interventions in Nairobi, and 3 specific interventions in Gem); and (d) Invited and accessed the "complete" primary package (all 5 primary interventions in Nairobi, and 4 or 5 interventions in Gem). In Gem, the partial package comprised three specific interventions: HIV testing services; school or community-based HIV and violence prevention; and social asset building. Rather than combining all individuals accessing multiple interventions into one group, this current categorization allowed us to explore whether a smaller set of interventions was also effective in influencing outcomes, which could indicate efficient programming options where resources are limited. This particular group of three interventions was the most common combination accessed by AGYW in Gem, reflecting what was prioritised by, and most feasible for, implementers in this setting. We distinguished ≥3 interventions from 1-2 interventions to align with the UNAIDS recommendation for combination prevention that integrates behavioral, biomedical, and structural interventions. We focussed on assessing whether a "complete package" was better than a "partial package" that was meaningful (that might be delivered as an alternative to a "complete package"), and we considered that if a partial package were designed/implemented then it should still include 3 or more interventions.

**Outcomes.** We considered 5 outcomes: 1) *Knowledge of HIV status,* defined as self-report of a HIV positive status or testing HIV negative in the previous 12 months; 2) *Condomless sex,* defined as absence of condoms at least once during a sexual encounter in the previous 12 months, among all AGYW and among those who were sexually active; 3) Number of lifetime sexual partners, grouped into ≥*1 lifetime partners* and ≥*2 lifetime partners;* 4) *Social support,* defined as a binary variable based on four questions relating to female networks and access to safe spaces: low social support (=0), defined as a "yes" to 0–2 questions; high social support (=1), defined as a "yes" to 3–4 questions [8]; and, 5) *Generalised self-efficacy* defined based on scores from 10 questions from the general self-efficacy scale measuring overall coping ability, and competence to solve problems and meet goals (high efficacy was defined as a score of ≥3.5 while a score of <3.5 represented low efficacy). Generalised self-efficacy refers to an individual's belief in their capacity to manage tasks, overcome challenges, and achieve goals. It is a key determinant of motivation, resilience, and problem-solving, and is relevant across multiple domains of life, including social well-being, education, and career development. Individuals with high self-efficacy are more likely to initiate coping strategies and maintain stability when facing adversity [8]. The selection of these particular outcomes was based on our conceptual framework, in which it was hypothesised that DREAMS interventions would influence the behavioural, biomedical, and social determinants of HIV risk [18]. Previously, we compared these outcomes between DREAMS invitees and non-invitees and did not distinguish among invitees in terms of the number of interventions that they actually received. All outcomes were based on endline follow-up data (2019) after three years of DREAMS implementation. Given that knowledge of HIV status can be influenced relatively quickly, with one HIV testing service within a year, we also analyzed knowledge of HIV status by 2018.

## Analysis

We used the same analytical approach that has been utilized in previous analyses in which we estimated the causal impact of DREAMS by comparing invitees with non-invitees [7,19]. First, we calculated descriptive summaries of the socio-demographic characteristics at cohort enrolment and outcomes in 2018 or 2019 by the four-category measure of exposure to DREAMS interventions, both overall and stratified by age group (13–17 and 18–22 years at cohort enrolment for Gem and 15–17 and 18–22 years at enrolment for Nairobi).

We implemented the analysis in stages. In the first stage, we used directed acyclic graphs (DAGs) to conceptualize the causal relationship between exposure to DREAMS interventions and outcomes [20] (S1 Text). Using the DAGs, we identified a set of potential confounding variables informed by how DREAMS was targeted, i.e., individual and household characteristics that were determinants of invitation to DREAMS and also could influence the outcomes [21]. The set of confounding variables that were controlled for in each analysis were: age group; measures of educational status, including highest educational achievement and current school enrolment; measures of household economic status, including a wealth index and food insecurity; marital status; sexual and pregnancy history; and orphanhood status. All confounding variables were as measured at cohort enrolment (2017 in Nairobi and 2018 in Gem).

In the second stage, we used multivariable logistic regression to summarize the association between DREAMS invitation and participation in interventions and each outcome. These models provided estimates of effect (odds ratios) that are conditional on covariates, and these are reported based on the unadjusted models, age- and site-adjusted models, as well as fully adjusted models that incorporated all the confounding variables identified from the DAGs. These analyses were done separately for each setting, and both overall and separately for younger and older AGYW.

In the third stage, we obtained marginal estimates of effect that are helpful for population-level inference, and comparisons were made in terms of risk differences in the proportion with the outcome. Using a causal inference framework, we assessed the impacts of exposure to DREAMS by comparing the expected percentage of AGYW with the outcome under three counterfactual scenarios: 1) that all AGYW were invited to DREAMS and accessed the complete package; 2) that all AGYW were invited to DREAMS and accessed the partial package; 3) that no AGYW were invited to DREAMS (the causal assumptions are summarised in S2 Text). The primary analysis used propensity-score regression adjustment to control for

confounding. We constructed a propensity score model in which the outcome was invitation to DREAMS by 2018 (yes or no), and the explanatory variables were the minimal set of confounding variables as identified by the DAG. By incorporating these explanatory variables into the propensity score model, we aimed (in the next step of the analysis, with regression adjustment for the propensity score) to balance the exposure groups on observed confounders and thereby limit confounding bias.

To predict the probability of an outcome (for example, knowledge of HIV status) in the scenario that all AGYW received the complete package, we fitted a logistic regression model with age group and the propensity score as the explanatory variables, with restriction to AGYW who were DREAMS invitees and accessed the complete package. From this model, we predicted the probability of the outcome for all AGYW who were followed up in 2019 (all outcomes) or 2018 (knowledge of HIV status outcome only) under this scenario. The average value of these probabilities was used to estimate the percentage of AGYW with the outcome under the counterfactual scenario that all AGYW were DREAMS invitees and accessed the complete package. We repeated this approach for AGYW who were DREAMS invitees and accessed the partial package, to estimate the percentage of AGYW with the outcome under the counterfactual scenario that all AGYW were DREAMS invitees and accessed the partial package. We also repeated this approach for AGYW who were not invited to DREAMS, to estimate the percentage of AGYW with the outcome under the counterfactual scenario that no AGYW were DREAMS invitees. We present these average predictions overall, and separately for younger and older AGYW in each setting.

Bootstrapping, using 1000 samples drawn with replacement, was applied to obtain confidence intervals for the predicted percentages with the outcome, and for the difference in the percentages between the three counter-factual scenarios. We also conducted sensitivity analyses, using inverse-probability-of-treatment weighting (with probability of treatment equal to the propensity score for invitees, and 1 minus the propensity score for non-invitees), and using predictions derived from a multivariable logistic regression model of the outcome variable adjusted for the minimal confounding set of explanatory variables.

### Ethics

Ethical approvals for the study were granted by research ethics committees at the London School of Hygiene & Tropical Medicine, Amref Health Africa and the Kenyan Medical Research Institute. Informed consent was obtained from participants aged 18 and older. For legal minors under 18 years, guardian consent was obtained first, followed by the girl's assent.

## Results

### Cohort enrolment and retention

A total of 1081 AGYW were enrolled into the cohort in Nairobi (urban setting) in 2017, and 1171 in Gem (rural setting) in 2018. The cohort retention in 2019 was high, at 79% (n = 852) in Nairobi and 87% (n = 1018) in Gem. Retention was at least 65% across most participant characteristics, as measured at enrolment. Retention was higher among those in- versus out-of-schooling at enrolment in Nairobi. In both Nairobi and Gem, retention was higher among those who had never had sex, compared to those who, at enrolment, had ever had sex [7].

### Exposure to DREAMS and uptake of the primary interventions

Among AGYW followed up in 2019 in Nairobi, 26% (n = 223 out of 852) had been invited to DREAMS and accessed the complete package of 5 interventions; 32% (n = 272) had been invited to DREAMS and accessed a partial package of 3–4 interventions; 16% (n = 133) had been invited to DREAMS and accessed <3 interventions; and 26% (n = 224) had not been invited to DREAMS by 2018 (Fig 2). Among AGYW followed up in 2019 in Gem, 24% (n = 243 out of 1018) had been

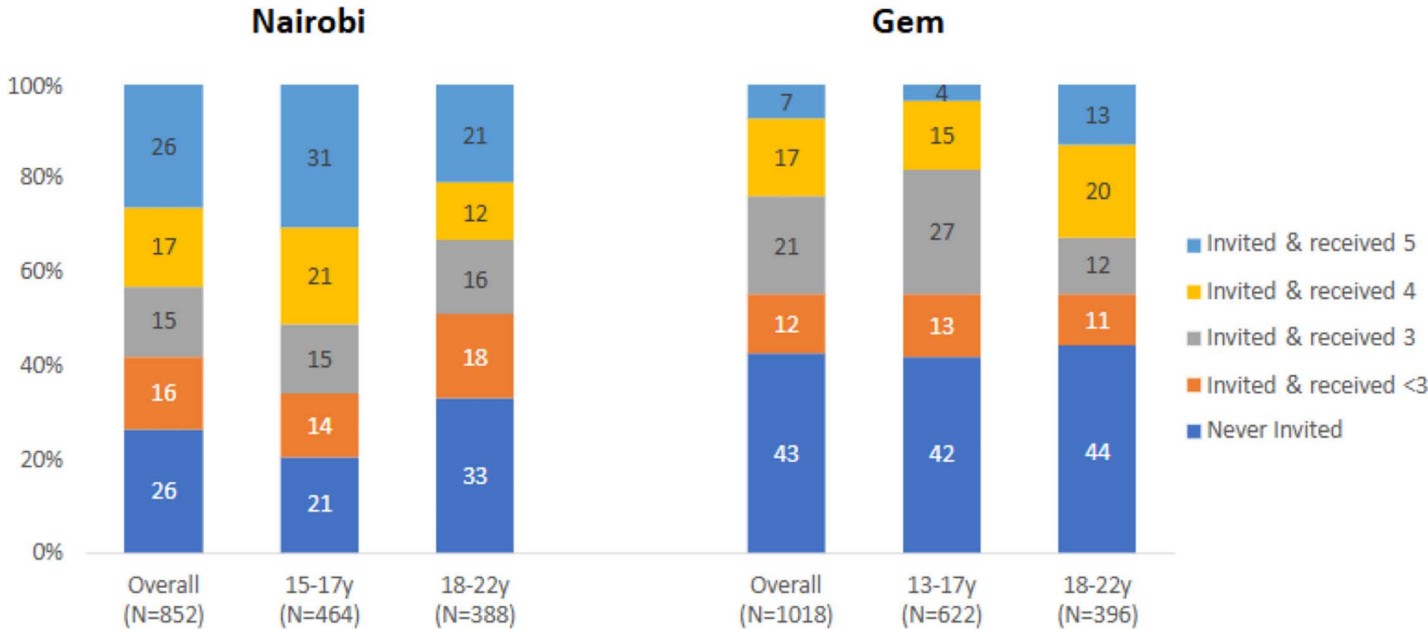

**Fig 2. Number of primary interventions accessed, overall and by age.**

invited to DREAMS and accessed the complete package of 4–5 interventions, 21% (n = 213) had been invited to DREAMS and accessed a partial package of 3 interventions; 12% (n = 126) had been invited to DREAMS and accessed <3 interventions; and 43% (n = 436) had not been invited to DREAMS by 2018 (Fig 2).

In Nairobi, AGYW invited to DREAMS and accessing either the partial or the complete packages were more likely to be younger, enrolled in school, to report household food insecurity, and less likely to report they had ever had sex or were ever married than those not invited to DREAMS (Table 1, S1 Table). The largest differentials were observed when comparing the AGYW who received the complete package with those not invited to DREAMS according to educational and sexual experience. For instance, in Nairobi, the percentage in school was 30% higher among AGYW accessing the complete package compared to those not invited to DREAMS.

In Gem, AGYW invited to DREAMS and accessing either the partial or the complete packages were more likely to be younger and to report household food insecurity, and less likely to report they had ever had sex, than those not invited to DREAMS (Table 2, S1 Table). The largest differentials were observed in education and household food insecurity and sexual experience when comparing AGYW who received the complete package with those not invited to DREAMS. For instance, in Gem, the percentage reporting household food insecurity was 15% higher among AGYW accessing the complete package compared to those not invited to DREAMS. Description of cohort characteristics by age are displayed in S2 Table.

### Estimated impact of DREAMS on outcomes

**Knowledge of HIV status 2018 & 2019.** In 2018, in both settings, knowledge of HIV status was higher among those accessing DREAMS multi-component packages (either partial or the complete packages), compared with those not invited to DREAMS. In Nairobi overall, knowledge of HIV status in 2018 was 88% among those accessing the partial package, and 92% among those accessing the complete package, compared to 61% among non-invitees. The corresponding percentages in Gem were 86% for the partial package and 83% for the complete package compared to 68% among non-invitees. Similar patterns were observed among the younger and older AGYW (Table 3).

**Table 1. Characteristics at enrolment, Nairobi.**

| | | Invited to DREAMS | | | |
|---|---|---|---|---|---|
| | | No | Yes | | |
| | Total (N = 852) | Never invited (N = 224) | Invited & accessed <3 primary interventions (N = 133) | Invited & accessed the partial package (N = 272) | Invited & accessed the complete package (N = 223) |
| | n (%) | n (%) | n (%) | n (%) | n (%) |
| **Age group** | | | | | |
| 15-17 | 464 (54.5) | 95 (42.4) | 64 (48.1) | 163 (59.9) | 142 (63.7) |
| 18-22 | 388 (45.5) | 129 (57.6) | 69 (51.9) | 109 (40.1) | 81 (36.3) |
| **Study site** | | | | | |
| Korogocho | 513 (60.2) | 143 (63.8) | 73 (54.9) | 174 (64.0) | 123 (55.2) |
| Viwandani | 339 (39.8) | 81 (36.2) | 60 (45.1) | 98 (36.0) | 100 (44.8) |
| **Currently in school** | | | | | |
| No | 312 (36.6) | 109 (48.7) | 74 (55.6) | 85 (31.3) | 44 (19.7) |
| Yes | 540 (63.4) | 115 (51.3) | 59 (44.4) | 187 (68.8) | 179 (80.3) |
| **Highest education level completed** | | | | | |
| None/incomplete primary | 92 (10.8) | 30 (13.4) | 16 (12.0) | 28 (10.3) | 18 (8.1) |
| Complete primary | 170 (20.0) | 54 (24.1) | 30 (22.6) | 46 (16.9) | 40 (17.9) |
| Some secondary | 410 (48.1) | 76 (33.9) | 54 (40.6) | 143 (52.6) | 137 (61.4) |
| Complete secondary/Tertiary | 180 (21.1) | 64 (28.6) | 33 (24.8) | 55 (20.2) | 28 (12.6) |
| **Sexual and pregnancy history** | | | | | |
| Never had sex | 557 (65.4) | 125 (55.8) | 69 (51.9) | 185 (68.0) | 178 (79.8) |
| Ever sex, never pregnant | 90 (10.6) | 26 (11.6) | 20 (15.0) | 26 (9.6) | 18 (8.1) |
| Ever pregnant | 205 (24.1) | 73 (32.6) | 44 (33.1) | 61 (22.4) | 27 (12.1) |
| Marital status | | | | | |
| Never married | 695 (81.6) | 161 (71.9) | 96 (72.2) | 232 (85.3) | 206 (92.4) |
| Ever married or living w partner | 157 (18.4) | 63 (28.1) | 37 (27.8) | 40 (14.7) | 17 (7.6) |
| **Orphanhood status** | | | | | |
| Not an orphan | 663 (77.8) | 170 (75.9) | 94 (70.7) | 218 (80.1) | 181 (81.2) |
| Single/double orphan | 189 (22.2) | 54 (24.1) | 39 (29.3) | 54 (19.9) | 42 (18.8) |
| **Food insecurity** | | | | | |
| No | 564 (66.2) | 166 (74.1) | 103 (77.4) | 166 (61.0) | 129 (57.8) |
| Yes | 288 (33.8) | 58 (25.9) | 30 (22.6) | 106 (39.0) | 94 (42.2) |
| **Self-assessed household poverty** | | | | | |
| Very poor | 115 (13.5) | 23 (10.3) | 8 (6.0) | 47 (17.3) | 37 (16.6) |
| Moderately poor | 672 (78.9) | 180 (80.4) | 112 (84.2) | 207 (76.1) | 173 (77.6) |
| Not poor | 65 (7.6) | 21 (9.4) | 13 (9.8) | 18 (6.6) | 13 (5.8) |
| **Wealth quantile** | | | | | |
| Poor | 303 (35.6) | 77 (34.4) | 39 (29.3) | 110 (40.4) | 77 (34.5) |
| Medium | 277 (32.5) | 79 (35.3) | 35 (26.3) | 83 (30.5) | 80 (35.9) |
| Wealthy | 272 (31.9) | 68 (30.4) | 59 (44.4) | 79 (29.0) | 66 (29.6) |
| **Ethnicity** | | | | | |
| Somali | 76 (8.9) | 16 (7.1) | 13 (9.8) | 23 (8.5) | 24 (10.8) |
| Kamba | 149 (17.5) | 40 (17.9) | 31 (23.3) | 37 (13.6) | 41 (18.4) |
| Kikuyu | 272 (31.9) | 61 (27.2) | 34 (25.6) | 101 (37.1) | 76 (34.1) |
| Kisii | 33 (3.9) | 11 (4.9) | 6 (4.5) | 11 (4.0) | 5 (2.2) |
| Luhya | 135 (15.8) | 36 (16.1) | 17 (12.8) | 48 (17.6) | 34 (15.2) |
| Luo | 134 (15.7) | 39 (17.4) | 25 (18.8) | 39 (14.3) | 31 (13.9) |
| Other | 53 (6.2) | 21 (9.4) | 7 (5.3) | 13 (4.8) | 12 (5.4) |

**Table 2. Characteristics at enrolment, Gem.**

| | | Invited to DREAMS | | | |
| | | No | Yes | | |
| | Total (N = 1018) | Never invited (N = 436) | Invited & accessed < 3 primary interventions (N = 126) | Invited & accessed the partial package (N = 213) | Invited & accessed the complete package (N = 243) |
| | N(%) | N(%) | N(%) | N(%) | N(%) |
| **Age group** | | | | | |
| 13-17 | 622(61.1) | 261(59.9) | 82(65.1) | 165(77.5) | 114(46.9) |
| 18-22 | 396(38.9) | 175(40.1) | 44(34.9) | 48(22.5) | 129(53.1) |
| **Educational attainment** | | | | | |
| Primary/None | 435(42.7) | 175(40.1) | 68(54.0) | 110(51.6) | 82(33.7) |
| Secondary and above | 372(36.5) | 143(32.8) | 38(30.2) | 70(32.9) | 121(49.8) |
| Unknown | 211(20.7) | 118(27.1) | 20(15.9) | 33(15.5) | 40(16.5) |
| **Sexual and pregnancy history** | | | | | |
| Never had sex | 701(68.9) | 279(64.0) | 80(63.5) | 183(85.9) | 159(65.4) |
| Ever sex, never pregnant | 158(15.5) | 76(17.4) | 26(20.6) | 21(9.9) | 35(14.4) |
| Ever pregnant | 159(15.6) | 81(18.6) | 20(15.9) | 9(4.2) | 49(20.2) |
| **Orphanhood** | | | | | |
| No | 615(60.4) | 259(59.4) | 80(63.5) | 133(62.4) | 143(58.8) |
| Maternal | 35(3.4) | 15(3.4) | 8(6.3) | 5(2.3) | 7(2.9) |
| Paternal | 92(9.0) | 36(8.3) | 8(6.3) | 20(9.4) | 28(11.5) |
| Total | 33(3.2) | 19(4.4) | 7(5.6) | 4(1.9) | 3(1.2) |
| Unknown | 243(23.9) | 107(24.5) | 23(18.3) | 51(23.9) | 62(25.5) |
| **Food insecurity** | | | | | |
| No | 789(77.5) | 360(82.6) | 102(81.0) | 163(76.5) | 164(67.5) |
| Yes | 229(22.5) | 76(17.4) | 24(19.0) | 50(23.5) | 79(32.5) |
| **Socio-economic status** | | | | | |
| Low | 424(41.7) | 157(36.0) | 54(42.9) | 85(39.9) | 128(52.7) |
| Middle | 195(19.2) | 83(19.0) | 27(21.4) | 49(23.0) | 36(14.8) |
| High | 399(39.2) | 196(45.0) | 45(35.7) | 79(37.1) | 79(32.5) |
| **Self-assessed poverty of household** | | | | | |
| Very Poor | 129(12.7) | 48(11.0) | 19(15.1) | 25(11.7) | 37(15.2) |
| Moderately poor | 731(71.8) | 307(70.4) | 93(73.8) | 153(71.8) | 178(73.3) |
| Not Poor | 158(15.5) | 81(18.6) | 14(11.1) | 35(16.4) | 28(11.5) |

Adjusted odds ratios (from multivariable logistic regression analysis) were consistent with these patterns, and in Nairobi, compared to non-invitees, knowledge of HIV status was higher among AGYW who received the complete package (aOR=12.20, 95%CI: 6.20–23.78) and those who received the partial package (aOR =7.19, 95%CI: 4.36–11.86). Similarly, in Gem, compared to non-invitees, knowledge of HIV status was higher among those who received the complete package (aOR =1.99, 95%CI: 1.23–3.23) and those who received the partial package (aOR =3.00, 95%CI: 1.92–4.68) (Table 3).

In the third stage of our analyses, we conducted causal inference analyses, which controlled for confounding using propensity scores and estimated the percentage with the outcome for different counterfactual scenarios. In Nairobi we estimated that the percentages of AGYW who would know their HIV status in 2018, comparing the scenarios that all accessed the partial package vs. none were invited, were 88% vs. 56% (a 32.0 percentage-point increase due to DREAMS with 95% confidence interval (95% CI) 23.8 to 40.2). The percentages comparing the scenarios that all accessed the complete

PLOS Global Public Health

**Table 3. Impact of DREAMS on outcomes, from multivariable logistic regression.**

| Outcome | Setting | | Exposure to DREAMS* | n/N | % | Unadjusted OR (95% CI) | Age & site adjusted OR (95% CI) | Multivariable adjusted OR (95% CI) | p-value |
|---|---|---|---|---|---|---|---|---|---|
| **Knowledge of HIV status, 2018** | Nairobi | Overall | Not invited | 129/212 | 60.8 | 1 | 1 | 1 | |
| | | | Accessed <3 interventions | 121/164 | 73.8 | 1.81 (1.16–2.82) | 1.98 (1.26–3.13) | 2.06 (1.27–3.32) | <0.001 |
| | | | Accessed partial package | 257/291 | 88.3 | 4.86 (3.10–7.64) | 6.04 (3.77–9.66) | 7.19 (4.36–11.86) | |
| | | | Accessed complete package | 155/169 | 91.7 | 7.12 (3.86–13.14) | 8.75 (4.66–16.46) | 12.15 (6.2–23.78) | |
| | Gem | Overall | Not invited | 351/514 | 68.3 | 1 | 1 | 1 | |
| | | | Accessed <3 interventions | 221/295 | 74.9 | 1.39 (1.00–1.91) | 1.47 (1.06–2.04) | 1.41 (1.01–1.97) | <0.001 |
| | | | Accessed partial package | 184/213 | 86.4 | 2.95 (1.91–4.54) | 3.11 (2.01–4.81) | 3.00 (1.92–4.68) | |
| | | | Accessed complete package | 124/149 | 83.2 | 2.30 (1.44–3.68) | 2.15 (1.34–3.45) | 1.99 (1.23–3.23) | |
| **Knowledge of HIV status, 2019** | Nairobi | Overall | Not invited | 144/224 | 64.3 | 1 | 1 | 1 | |
| | | | Accessed <3 interventions | 97/133 | 72.9 | 1.50 (0.94–2.40) | 1.51 (0.94–2.44) | 1.53 (0.93–2.51) | <0.001 |
| | | | Accessed partial package | 224/272 | 82.4 | 2.59 (1.71–3.92) | 2.93 (1.91–4.48) | 2.81 (1.80–4.38) | |
| | | | Accessed complete package | 197/223 | 88.3 | 4.21 (2.57–6.88) | 4.98 (3.00–8.26) | 4.85 (2.85–8.24) | |
| | Gem | Overall | Not invited | 360/436 | 82.6 | 1 | 1 | 1 | |
| | | | Accessed <3 interventions | 108/126 | 85.7 | 1.27 (0.73–2.21) | 1.26 (0.72–2.20) | 1.24 (0.70–2.20) | 0.009 |
| | | | Accessed partial package | 185/213 | 86.9 | 1.39 (0.87–2.23) | 1.37 (0.86–2.21) | 1.38 (0.85–2.24) | |
| | | | Accessed complete package | 223/243 | 91.8 | 2.35 (1.40–3.96) | 2.38 (1.41–4.01) | 2.41 (1.41–4.13) | |
| **Condomless sex, 2019** | Nairobi | Overall | Not invited | 91/224 | 40.6 | 1 | 1 | 1 | |
| | | | Accessed <3 interventions | 55/133 | 41.4 | 1.00 (0.67–1.59) | 1.09 (0.67–1.77) | 0.91 (0.51–1.61) | 0.473 |
| | | | Accessed partial package | 93/272 | 34.2 | 0.76 (0.53–1.09) | 1.07 (0.71–1.62) | 1.17 (0.72–1.90) | |
| | | | Accessed complete package | 49/223 | 22.0 | 0.41 (0.27–0.62) | 0.58 (0.37–0.91) | 0.80 (0.46–1.36) | |
| | Gem | Overall | Not invited | 82/436 | 18.8 | 1 | 1 | 1 | <0.001 |
| | | | Accessed <3 interventions | 26/126 | 20.6 | 1.12 (0.69–1.84) | 1.31 (0.76–2.25) | 1.29 (0.73–2.26) | |
| | | | Accessed partial package | 10/213 | 4.7 | 0.21 (0.11–0.42) | 0.28 (0.14–0.57) | 0.26 (0.13–0.54) | |
| | | | Accessed complete package | 29/243 | 11.9 | 0.59 (0.37–0.92) | 0.42 (0.26–0.68) | 0.44 (0.26–0.73) | |
| **Condomless sex among sexually active AGYW, 2019** | Nairobi | Overall | Not invited | 91/101 | 90.1 | 1 | 1 | 1 | |
| | | | Accessed <3 interventions | 55/66 | 83.3 | 0.55 (0.22–1.38) | 0.59 (0.23–1.49) | 0.45 (0.16–1.27) | 0.128 |
| | | | Accessed partial package | 93/111 | 83.8 | 0.57 (0.25–1.30) | 0.66 (0.28–1.53) | 0.75 (0.29–1.91) | |
| | | | Accessed complete package | 49/60 | 81.7 | 0.49 (0.19–1.23) | 0.57 (0.22–1.47) | 0.77 (0.27–2.21) | |
| | Gem | Overall | Not invited | 82/153 | 53.6 | 1 | 1 | 1 | |
| | | | Accessed <3 interventions | 26/41 | 63.4 | 1.50 (0.74–3.05) | 1.57 (0.77–3.22) | 1.62 (0.75–3.47) | 0.020 |
| | | | Accessed partial package | 10/25 | 40.0 | 0.58 (0.24–1.37) | 0.61 (0.26–1.45) | 0.71 (0.28–1.80) | |
| | | | Accessed complete package | 29/81 | 35.8 | 0.48 (0.28–0.84) | 0.47 (0.27–0.82) | 0.49 (0.27–0.89) | |
| **≥1 lifetime partner, 2019** | Nairobi | Overall | Not invited | 128/224 | 57.1 | 1 | 1 | 1 | |
| | | | Accessed <3 interventions | 84/133 | 63.2 | 1.29 (0.83–2.00) | 1.54 (0.93–2.57) | 1.31 (0.75–2.28) | 0.024 |
| | | | Accessed partial package | 146/272 | 53.7 | 0.87 (0.61–1.24) | 1.33 (0.88–2.01) | 1.56 (0.99–2.45) | |
| | | | Accessed complete package | 84/223 | 37.7 | 0.45 (0.31–0.66) | 0.64 (0.41–0.99) | 0.83 (0.51–1.35) | |
| | Gem | Overall | Not invited | 183/436 | 42.0 | 1 | 1 | 1 | |
| | | | Accessed <3 interventions | 49/126 | 38.9 | 0.88 (0.59–1.32) | 0.99 (0.61–1.59) | 0.99 (0.61–1.61) | <0.001 |
| | | | Accessed partial package | 35/213 | 16.4 | 0.27 (0.18–0.41) | 0.31 (0.20–0.50) | 0.32 (0.20–0.52) | |
| | | | Accessed complete package | 105/243 | 43.2 | 1.05 (0.77–1.44) | 0.72 (0.49–1.06) | 0.78 (0.52–1.15) | |

*(Continued)*

**Table 3.** (Continued)

| Outcome | Setting | | Exposure to DREAMS* | n/N | % | Unadjusted OR (95% CI) | Age & site adjusted OR (95% CI) | Multivariable adjusted OR (95% CI) | p-value |
|---|---|---|---|---|---|---|---|---|---|
| ≥2 lifetime partner, 2019 | Nairobi | Overall | Not invited | 67/224 | 29.9 | 1 | 1 | 1 | |
| | | | Accessed <3 interventions | 37/133 | 27.8 | 0.90 (0.56–1.45) | 1.00 (0.58–1.61) | 0.91 (0.53–1.55) | 0.166 |
| | | | Accessed partial package | 63/272 | 23.2 | 0.71 (0.47–1.06) | 0.93 (0.61–1.44) | 1.00 (0.62–1.57) | |
| | | | Accessed complete package | 31/223 | 13.9 | 0.38 (0.24–0.61) | 0.52 (0.31–0.85) | 0.58 (0.34–1.00) | |
| | Gem | Overall | Not invited | 102/436 | 23.4 | 1 | 1 | 1 | |
| | | | Accessed <3 interventions | 28/126 | 22.2 | 0.94 (0.58–1.50) | 1.07 (0.63–1.85) | 1.11 (0.63–1.95) | <0.001 |
| | | | Accessed partial package | 12/213 | 5.6 | 0.20 (0.10–0.36) | 0.25 (0.13–0.48) | 0.25 (0.13–0.49) | |
| | | | Accessed complete package | 44/243 | 18.1 | 0.72 (0.49–1.07) | 0.49 (0.32–0.76) | 0.55 (0.35–0.88) | |
| Social support, 2019 | Nairobi | Overall | Not invited | 111/224 | 49.6 | 1 | 1 | 1 | |
| | | | Accessed <3 interventions | 74/133 | 55.6 | 1.28 (0.83–1.96) | 1.21 (0.79–1.88) | 1.20 (0.76–1.88) | 0.015 |
| | | | Accessed partial package | 148/272 | 54.4 | 1.22 (0.85–1.73) | 1.22 (0.85–1.74) | 1.23 (0.84–1.79) | |
| | | | Accessed complete package | 147/223 | 65.9 | 1.97 (1.35–2.88) | 1.89 (1.28–2.79) | 1.92 (1.28–2.9) | |
| | Gem | Overall | Not invited | 122/436 | 28.0 | 1 | 1 | 1 | |
| | | | Accessed <3 interventions | 38/126 | 30.2 | 1.11 (0.72–1.72) | 1.12 (0.72–1.73) | 1.12 (0.72–1.75) | <0.001 |
| | | | Accessed partial package | 76/213 | 35.7 | 1.43 (1.01–2.03) | 1.46 (1.03–2.08) | 1.62 (1.13–2.34) | |
| | | | Accessed complete package | 133/243 | 54.7 | 3.11 (2.24–4.32) | 3.06 (2.20–4.26) | 3.29 (2.33–4.66) | |
| | | | Accessed complete package | 73/129 | 56.6 | 3.17 (1.97–5.11) | 3.17 (1.97–5.11) | 3.54 (2.11–5.94) | |
| Generalised self-efficacy, 2019 | Nairobi | Overall | Not invited | 113/224 | 50.4 | 1 | 1 | 1 | |
| | | | Accessed <3 interventions | 71/133 | 53.4 | 1.12 (0.73–1.73) | 1.11 (0.72–1.71) | 1.08 (0.69–1.68) | |
| | | | Accessed partial package | 149/272 | 54.8 | 1.19 (0.83–1.70) | 1.22 (0.85–1.75) | 1.21 (0.84–1.76) | 0.203 |
| | | | Accessed complete package | 132/223 | 59.2 | 1.42 (0.98–2.07) | 1.44 (0.98–2.10) | 1.53 (1.02–2.28) | |
| | Gem | Overall | Not invited | 145/436 | 33.3 | 1 | 1 | 1 | |
| | | | Accessed <3 interventions | 43/126 | 34.1 | 1.04 (0.68–1.58) | 1.07 (0.70–1.63) | 1.21 (0.78–1.88) | 0.115 |
| | | | Accessed partial package | 80/213 | 37.6 | 1.21 (0.86–1.70) | 1.34 (0.94–1.89) | 1.50 (1.04–2.16) | |
| | | | Accessed complete package | 83/243 | 34.2 | 1.04 (0.75–1.45) | 0.97 (0.69–1.36) | 0.97 (0.68–1.38) | |

In Nairobi: partial package is 3–4 interventions, and complete package is 5 interventions; in Gem partial package is 3 specific interventions, complete package is 4 or 5 interventions. p value – Likelihood ratio (LRT) test, an overall test of whether there are differences in the proportion with the outcome among the 4 categories that are being compared.

package vs. none were invited, were 91% vs. 56% (+34.8 percentage-points with 95% CI 27.0 to 43.2). In Gem, the corresponding percentages comparing the scenarios that all accessed the partial package vs. none were invited, were 84% vs. 69% (+ 15.7 percentage-points with 95% CI 8.6 to 22.4), and comparing the scenarios that all accessed the complete package vs. none were invited, were 82% vs. 69% (+13.9 percentage-points with 95% CI 5.5 to 21.4). Differentials were larger among younger than older AGYW (Table 4).

In 2019, the differentials in knowledge of HIV status between the three scenarios were smaller than in 2018 in both settings, though the overall patterns were consistent with 2018. In Gem, however, statistical evidence of an increase was only found under the scenario that all AGYW accessed the complete package (Table 4).

**Condomless sex at least once in the previous 12 months in 2019.** The percentages who reported condomless sex were lower among those accessing the partial and the complete packages compared to non-invitees. In Nairobi

**Table 4. Estimated impact of DREAMS on outcomes, comparing the counterfactual scenarios that all AGYW accessed the partial package or all AGYW accessed the complete package vs. no AGYW were invited to DREAMS by 2018.**

| Outcome | Setting and age group | | % with outcome in total study population (observed) | Estimated % with outcome if no AGYW are invited to DREAMS, & 95% CI (A) | Estimated % with outcome if all AGYW access the partial package, & 95% CI (B) | Estimated % with outcome if all AGYW access the complete package, & 95% CI (C) | Difference in estimated % with outcome: all AGYW access partial package (B) – no AGYW invited to DREAMS (A), & 95% CI | Difference in estimated % with outcome: all AGYW access complete package (C) – no AGYW invited to DREAMS (A), & 95% CI |
|---|---|---|---|---|---|---|---|---|
| Knowledge of HIV status, 2018 | Nairobi | Overall | 79.2 | 56.2 (49.1–63.7) | 88.2 (84.4–91.8) | 91.0 (85.8–95.2) | 32.0 (23.8–40.2) | 34.8 (27.0–43.2) |
| | | 15-17y | 75.5 | 44.7 (34.6–54.7) | 88.7 (83.7–93.3) | 91.0 (85.8–95.8) | 44.0 (32.4–55.3) | 46.3 (35.0–57.8) |
| | | 18-22y | 83.8 | 70.9 (62.4–78.9) | 87.6 (80.8–93.3) | 90.8 (78.3–98.0) | 16.8 (6.4–26.9) | 19.9 (5.4–31.4) |
| | Gem | Overall | 75.1 | 68.5 (64.5–72.8) | 84.2 (78.3–90.1) | 82.4 (75.0–88.9) | 15.7 (8.6–22.4) | 13.9 (5.5–21.4) |
| | | 13-17y | 71.9 | 64.3 (59.2–70.2) | 86.4 (80.3–91.9) | 79.7 (68.2–89.2) | 22.1 (13.0–29.4) | 15.4 (2.1–26.3) |
| | | 18-22y | 79.7 | 74.4 (68.7–80.1) | 81.2 (70.5–91.7) | 86.2 (79.2–92.8) | 6.8 (-6.0–18.5) | 11.8 (2.4–20.4) |
| Knowledge of HIV status, 2019 | Nairobi | Overall | 77.7 | 62.8 (56.1–69.4) | 82.4 (77.7–86.7) | 87.6 (82.4–91.9) | 19.6 (10.7–27.0) | 24.8 (16.4–32.6) |
| | | 15-17y | 77.2 | 57.5 (47.9–67.6) | 81.3 (75.1–87.0) | 90.4 (85.0–94.9) | 23.8 (11.7–34.7) | 32.8 (21.3–44.1) |
| | | 18-22y | 78.4 | 69.1 (60.9–77.6) | 83.8 (76.4–90.4) | 84.2 (75.1–91.7) | 14.7 (2.8–25.2) | 15.1 (3.3–27.1) |
| | Gem | Overall | 86.1 | 82.3 (78.5–86.2) | 85.7 (80.0–91.0) | 92.4 (88.3–95.8) | 3.4 (-3.5–9.8) | 10.0 (4.5–15.2) |
| | | 13-17y | 86.2 | 82.6 (78.0–87.1) | 86.8 (81.2–92.0) | 93.2 (88.1–97.3) | 4.2 (-3.3–11.2) | 10.6 (3.9–17.1) |
| | | 18-22y | 85.9 | 82.0 (76.0–88.0) | 84.0 (72.7–94.1) | 91.0 (85.6–95.9) | 2.0 (-10.7–14.0) | 9.1 (0.7–16.8) |
| Condomless sex, 2019 | Nairobi | Overall | 33.8 | 31.6 (26.2–37.1) | 36.2 (30.4–41.7) | 26.4 (20.2–33.1) | 4.6 (-3.0–11.9) | -5.2 (-13.3–3.1) |
| | | 15-17y | 17.7 | 11.0 (5.4–17.6) | 22.5 (16.1–29.1) | 13.5 (7.9–19.7) | 11.5 (2.5–20.0) | 2.5 (-5.9–10.5) |
| | | 18-22y | 53.1 | 56.1 (47.3–64.8) | 52.5 (43.5–61.7) | 41.8 (29.7–53.6) | -3.6 (-16.4–8.9) | -14.4 (-28.1–0.2) |
| | Gem | Overall | 14.4 | 18.7 (15.1–22.9) | 6.4 (2.7–11.4) | 9.5 (6.6–13.4) | -12.2 (-17–6.4) | -9.1 (-13.6–4.1) |
| | | 13-17y | 4.8 | 6.6 (3.8–9.4) | 2.5 (0.6–5.0) | 0.9 (0.8–3.3) | -4.1 (-7.6–0.2) | -5.7 (-8.1–-1.7) |
| | | 18-22y | 29.5 | 37.7 (29.6–45.3) | 12.7 (4.0–23.5) | 23.0 (15.3–30.8) | -25.0 (-36.6–13) | -14.6 (-25.3–4.5) |
| Condomless sex among sexually active AGYW, 2019 | Nairobi | Overall | 85.2 | 82.0 (68.0–89.7) | 80.5 (68.4–89.9) | 84.3 (77.3–91.0) | -1.5 (-16.0–15.8) | 2.3 (-8.1–17.1) |
| | | 15-17y | 80.4 | 57.7 (31.3–82.0) | 72.5 (49.9–91.6) | 86.5 (75.2–95.6) | 14.8 (-17.2–47.4) | 28.7 (1.9–59.0) |
| | | 18-22y | 87.3 | 92.5 (85.1–97.0) | 84.0 (71.8–93.8) | 83.3 (73.8–91.1) | -8.5 (-21.9–3.7) | -9.1 (-19.0–1.5) |
| | Gem | Overall | 49.0 | 51.8 (43.9–59.4) | 39.2 (19.1–57.9) | 34.0 (25.7–45.0) | -12.6 (-32.7–8.8) | -17.8 (-29.2–-2.8) |
| | | 13-17y | 41.1 | 46.9 (30.8–65.0) | 44.1 (10.9–76.6) | 6.6 (4.6–25.0) | -2.9 (-38.7–32.4) | -40.3 (-55.7–-13.8) |
| | | 18-22y | 51.5 | 53.4 (44.0–62.6) | 37.7 (13.3–60.0) | 42.8 (31–56.4) | -15.7 (-40.9–10.3) | -10.6 (-25.1–5.3) |
| ≥1 lifetime partners, 2019 | Nairobi | Overall | 51.9 | 48.0 (41.7–54.5) | 56.2 (50.1–61.5) | 44.0 (37.9–50.4) | 8.2 (-0.4–16.1) | -4.0 (-12.4–4.2) |
| | | 15-17y | 30.6 | 26.1 (17.7–36.0) | 35.3 (27.7–42.6) | 21.8 (15.6–28.8) | 9.2 (-2.6–20.9) | -4.3 (-15.3–7.1) |
| | | 18-22y | 77.3 | 74.1 (66.1–82.1) | 81.2 (73.6–88.6) | 70.5 (59.7–80.1) | 7.2 (-4.3–17.6) | -3.6 (-15.7–8.3) |
| | Gem | Overall | 36.5 | 41.6 (37.4–45.8) | 23.3 (16.5–29.7) | 39.5 (33.9–45.9) | -18.3 (-26.0–10.9) | -2.1 (-9.2–5.5) |
| | | 13-17y | 15.9 | 19.4 (14.5–24.1) | 8.2 (4.0–12.9) | 21.3 (14.1–29.5) | -11.1 (-17.4–4.9) | 2.0 (-6.7–11.3) |
| | | 18-22y | 68.9 | 76.5 (69.7–82.6) | 46.9 (31.6–61.1) | 68.1 (60.3–75.5) | -29.7 (-46.4–14.5) | -8.4 (-18.8–9.3) |
| ≥2 lifetime partners, 2019 | Nairobi | Overall | 23.2 | 25.7 (20.5–31.7) | 24.6 (19.4–30.0) | 17.7 (11.8–23.5) | -1.1 (-8.4–6.6) | -8.0 (-15.9–0.0) |
| | | 15-17y | 10.6 | 13.5 (6.8–21.0) | 14.0 (8.9–19.6) | 3.6 (0.8–7.5) | 0.5 (-8.5–9.5) | -9.9 (-17.7–2.4) |
| | | 18-22y | 38.4 | 40.3 (32.3–48.8) | 37.2 (28.1–47.2) | 34.4 (22.2–46.5) | -3.1 (-14.9–10.2) | -5.8 (-19.9–9.3) |
| | Gem | Overall | 18.3 | 23.3 (19.7–26.9) | 9.7 (5.9–34.6) | 15.7 (11.9–20.1) | -13.6 (-18.7–11.9) | -7.6 (-12.4–2.2) |
| | | 13-17y | 5.1 | 8.9 (5.8–12.3) | 0.6 (0.6–2.5) | 2.0 (0.9–5.3) | -8.3 (-11.2–-4.6) | -6.9 (-10.6–-2.0) |
| | | 18-22y | 38.9 | 45.9 (38.5–53.1) | 24.0 (11.8–37.4) | 37.1 (28.3–45.5) | -21.9 (-36.4–6.7) | -8.8 (-20.4–2.1) |

*(Continued)*

**Table 4.** (Continued)

| Outcome | Setting and age group | | % with outcome in total study population (observed) | Estimated % with outcome if no AGYW are invited to DREAMS, & 95% CI (A) | Estimated % with outcome if all AGYW access the partial package, & 95% CI (B) | Estimated % with outcome if all AGYW access the complete package, & 95% CI (C) | Difference in estimated % with outcome: all AGYW access partial package (B) – no AGYW invited to DREAMS (A), & 95% CI | Difference in estimated % with outcome: all AGYW access complete package (C) – no AGYW invited to DREAMS (A), & 95% CI |
|---|---|---|---|---|---|---|---|---|
| Social support, 2019 | Nairobi | Overall | 56.3 | 49.2 (42.0–56.6) | 53.8 (47.8–60.2) | 63.1 (56.2–70.2) | 4.5 (-4.4–13.9) | 13.9 (3.3–23.6) |
| | | 15-17y | 57.3 | 46.2 (35.7–56.2) | 54.2 (46.4–61.8) | 67.3 (59.6–75.4) | 8.0 (-4.1–20.3) | 21.2 (7.8–32.9) |
| | | 18-22y | 55.2 | 52.9 (44.6–61.7) | 53.3 (43.8–63.0) | 58.0 (47.0–69.7) | 0.4 (-12.3–13.7) | 5.1 (-8.5–19.4) |
| | Gem | Overall | 36.2 | 27.8 (23.4–32.5) | 35.4 (28.0–42.8) | 55.0 (47.9–61.8) | 7.7 (-0.5–16.2) | 27.2 (19.2–35.5) |
| | | 13-17y | 34.1 | 27.1 (21.8–33.2) | 36.5 (29.4–43.9) | 53.4 (43.8–62.5) | 9.5 (0.4–18.8) | 26.3 (16.2–37.1) |
| | | 18-22y | 39.6 | 28.9 (21.9–36.2) | 33.8 (19.5–49.2) | 57.6 (48.5–66.0) | 4.9 (-10.0–21.6) | 28.7 (16.9–39.6) |
| Generalised self-efficacy, 2019 | Nairobi | Overall | 54.6 | 50.9 (43.7–57.7) | 54.4 (48.0–60.0) | 61.2 (54.4–67.6) | 3.5 (-5.6–13.0) | 10.3 (0.5–20.4) |
| | | 15-17y | 53.2 | 50.9 (39.8–60.8) | 54.3 (46.6–61.8) | 55.9 (46.6–64.1) | 3.4 (-9.2–16.6) | 5.0 (-8.6–18.5) |
| | | 18-22y | 56.2 | 50.9 (42.1–59.8) | 54.5 (44.1–63.4) | 67.5 (57.0–77.7) | 3.6 (-9.1–16.8) | 16.6 (2.9–29.3) |
| | Gem | Overall | 34.5 | 31.8 (27.4–36.2) | 40.3 (33.1–47.4) | 32.7 (26.8–38.6) | 8.5 (-0.1–17.4) | 0.9 (-6.4–8.3) |
| | | 13-17y | 30.1 | 26.4 (21.2–31.8) | 35.7 (28.0–42.8) | 26.8 (18.8–35.2) | 9.3 (0.0–18.6) | 0.5 (-8.5–9.8) |
| | | 18-22y | 41.4 | 40.3 (33.0–48.1) | 47.4 (33.2–62.2) | 41.9 (33.4–50.5) | 7.1 (-9.6–23.8) | 1.6 (-10.5–13.2) |

| Colour codes | |
|---|---|
| Green: Positive effect, strong evidence | Amber: Positive effect, weak evidence | Gray: No effect |
| Red: Negative effect, strong evidence | Maroon: Negative effect, weak evidence | |

the percentages were 34% for the partial package and 22% for the complete package, vs 41% among non-invitees. Compared to non-invitees, condomless sex was lower among those who received the complete package, (aOR =0.80, 95%CI: 0.46–1.36). In Gem, the percentages were 5% for the partial package and 12% for the complete package, vs 19% among non-invitees. Compared to non-invitees, condomless sex was lower for those who received the partial package (aOR =0.26, 95%CI: 0.13–0.54) and for those who received the complete package (aOR=0.44, 95%CI: 0.26-0.73) (Table 3).

In causal analyses in which we estimated the percentage of AGYW with the outcome for different counterfactual scenarios, in Nairobi we found no evidence of impact of the complete package or partial package on condomless sex among all AGYW. Specifically, we estimated that the percentages of AGYW who would report condomless sex, comparing the scenarios that all accessed the partial package vs. none were invited, were 36% vs. 32% [difference: 4.6%, 95% CI: -3.5 to 9.8]; while the percentages comparing the scenarios that all accessed the complete package vs. none were invited, were 26% vs. 32% [difference: -5.2, 95% CI: -13.3 to 3.1)] (Table 4).

Among younger AGYW in Nairobi we estimated that the percentages of AGYW who would report condomless sex, comparing the scenarios that all accessed the partial package vs. none were invited, were 23% vs. 11%, with evidence of an increase in condomless sex [difference: 11.5, 95% CI: 2.5 to 20.0]. There was no evidence of a difference in condomless sex when we compared the scenarios that all accessed the complete package vs. none were invited: 14% vs. 11%, respectively [difference: 2.5, 95% CI: -5.9 to 10.5].

In Gem we found evidence of impact of the complete package and partial package on condomless sex among all AGYW. We estimated that the percentages comparing the scenarios that all accessed the partial package vs. none were invited, were 6% vs 19% [difference: -12.2, 95% CI: -17.0 to -6.4] and comparing the scenarios that all accessed the complete package vs. none were invited, were 10% vs 19% [difference: -9.1, 95% CI: -13.6 to -4.1)]. Among younger AGYW in

Gem, there was evidence of a reduction in condomless sex for both the partial [difference; -4.1, 95% CI: -7.6 to -0.2] and complete packages [difference: -5.7,95% CI: -8.1 to -1.7)] (Table 4).

Among sexually active older AGYW in Nairobi, there was weak evidence of a reduction in the percentages who would report condomless sex under the scenario that all received the partial package and in the scenario that all received the complete package, with an estimated difference of -8.5 percentage points [95% CI: -21.9 to 3.7] for the partial package, and an estimated difference of -9.1 percentage points [95% CI: -19.0 to 1.5] for the complete package. In Gem, there was no evidence of a differential among older AGYW for either of the packages, but there was evidence of a reduction among younger AGYW when comparing the scenario that all accessed the complete package vs. if none were invited [difference: -40.3, 95% CI: -55.7 to -13.8)] (Table 4).

**Lifetime partners, in 2019.** In Nairobi, the percentages who reported at least two lifetime partners were 30% for non-invitees, 23% for the partial package and 14% for the complete package; (aOR=0.58, 95%CI: 0.34–1.00) for the complete package compared to non-invitees (Table 3). In causal analyses in which we estimated the percentage of AGYW with the outcome for different counterfactual scenarios, overall there was evidence of a difference when comparing AGYW who received the complete package to non-invitees [difference: -8.0 with 95% CI: -15.9 to 0.0]. Among younger AGYW there was no evidence of a reduction in at least two lifetime partners when comparing AGYW who received the partial package with non-invitees [difference, 0.5 with 95% CI: -8.5 to 9.5] but there was evidence of a reduction comparing AGYW who received the complete package with non-invitees [difference: -9.9, 95% CI: -17.7 to -2.4] (Table 4).

In Gem, the percentages who reported at least two lifetime partners were 23% for non-invitees, 6% for the partial package and 18% for the complete package; (aOR=0.25, 95%CI: 0.13–0.49) for the partial package and (aOR=0.55, 95%CI: 0.35–0.88) for the complete package compared to non-invitees (Table 3). Comparing counterfactual scenarios, there was no statistical evidence of a reduction for AGYW who received the partial package [difference: -13.6, 95% CI: -18.7 to 11.9], but there was evidence of a reduction for AGYW who received the complete package [difference: -7.6, 95% CI: -12.4 to -2.2]. Among younger AGYW, there was evidence of reductions for both the partial package [difference: -8.3, 95% CI: -11.2 to -4.6] and complete package [difference: -6.9, 95% CI: -10.6 to -2.0]. Among older AGYW, there was evidence only of a reduction for the partial package [difference: -21.9, 95% CI: -36.4 to -6.7] (Table 4).

In Nairobi, in the causal analyses, there was weak evidence of an increase in the estimated percentages who reported at least one lifetime partner, comparing AGYW who received the partial package to non-invitees [difference: 8.2, 95% CI: -0.4 to 16.1], but no evidence of a difference when comparing those who received the complete package to non-invitees [difference: -4.0, 95% CI: -12.4 to 4.2]. In Gem, there was evidence of a reduction in the estimated percentages reporting at least one lifetime partner when comparing AGYW who received the partial package to non-invitees among all AGYW [difference, -18.3, 95% CI: -26.0 to -10.9] and in both age groups (younger AGYW:[difference, -11.1, 95% CI: -17.4 to -4.9]) (older AGYW:[difference, -29.7, 95% CI: -46.4 to -14.5]). However, the statistical evidence for a reduction was weak when comparing AGYW who received the complete package to non-invitees (Table 4).

**Social support.** The percentages who reported high levels of social support were higher among those accessing the partial and the complete packages compared to non-invitees in both settings: overall, in Nairobi, the percentages were 54% among those accessing the partial package and 66% among those accessing the complete package, vs 50% among non-invitees. In Gem, the corresponding percentages were 36% and 55%, vs 28% respectively. Compared to non-invitees, social support was higher for those who received the complete package (Nairobi aOR=1.92, 95%CI: 1.28–2.90; Gem aOR =3.29, 95%CI: 2.33–4.66), and for those who received the partial package in Gem (aOR =1.62, 95%CI: 1.13–2.34) (Table 3).

In Nairobi, we found evidence of an increase in the estimated percentage reporting high levels of social support for the complete package among all AGYW [63% vs. 49%, difference: 13.9, 95% CI: 3.3 to 23.6] and among younger AGYW [67% vs. 46%, difference: 21.2, 95% CI: 7.8 to 32.9] but not the partial package [54% vs. 46%, difference: 8.0, 95% CI: -4.1 to 20.3)]. Among older AGYW, there was no statistical evidence of an increase in levels of social support for either the partial or the complete packages (Table 4).

In Gem, we found evidence of an increase in social support for both packages among younger AGYW (for the partial package: [37% vs. 27%, difference: 9.5, 95% CI: 0.4 to 18.8]; and for the complete package: 53% vs. 27%, difference: 26.3, 95% CI: 16.2 to 37.1]). Among older AGYW in Gem, we found evidence of an increase for the complete package [58% vs 29%, difference: 28.7, 95% CI: 16.9 to 39.6], but not for the partial package [34% vs 29%, difference: 4.9, 95% CI: -10.0 to 21.6] (Table 4).

**Generalized self-efficacy.** Among younger AGYW, the overall percentage who reported high levels of generalised self-efficacy was 53% in Nairobi, with similar percentages across the exposure categories. In Gem the overall figure was 30%, with a higher percentage among those accessing the partial package (35%) than those accessing the complete package (26%) and the non-invitees (27%).

In Nairobi, in adjusted analyses generalised self-efficacy was higher among AGYW who received the complete package (aOR=1.53, 95%CI: 1.02–2.28) and those who received the partial package (aOR =1.21, 95%CI: 0.84-1.87) compared to non-invitees. In Gem, compared to non-invitees, generalised self-efficacy was higher for those who received the partial package (aOR =1.50, 95%CI: 1.04-2.16) but not for those who received the complete package (aOR =0.97, 95%CI: 0.68–1.38) (Table 3).

In Nairobi, we found no evidence of an increase in generalised self-efficacy among younger AGYW who received the partial or the complete packages (Table 4). In Gem, we found evidence of an increase in the percentage with high self-efficacy comparing scenarios that all accessed the partial package vs if none were invited [among younger AGYW: 36% vs. 26%, difference = 9.3, 95% CI: 0.0 to 18.6] but not when comparing the scenario that all accessed the complete package vs if none were invited [27% vs. 26%, difference: 0.5, 95% CI: -8.5 to 9.8] among younger AGYW (Table 4).

Among older AGYW, the percentages who reported high levels of generalised self-efficacy was 56% in Nairobi (55% among those accessing the partial package and 65% among those accessing the complete package vs. 50% among non-invitees) and 41% in Gem (with similar percentages across the exposure categories) (Table 3). In Nairobi, among older AGYW, we found evidence of an increase in the percentage with high self-efficacy comparing the scenarios that all accessed the complete package vs if none were invited [68% vs. 51%, difference: 16.6, 95% CI: 2.9 to 29.3], but not in the scenarios if all accessed the partial package vs if none were invited [55% vs. 51%, difference: 3.6, 95% CI: -9.1 to 16.8]. In Gem, among older AGYW, there was no evidence of an increase in self-efficacy for either of the packages (Table 4).

## Sensitivity analyses

For all outcomes, findings were similar in sensitivity analyses (S3–S10 Tables).

## Discussion

### Key findings

We compared the impact of different levels of exposure to DREAMS intervention packages (partial and complete packages of 'primary' DREAMS interventions) on several outcomes using representative cohorts of AGYW surveyed between 2017/18 and 2019 in urban (Nairobi) and rural (Gem) Kenya.

In Nairobi, we found evidence of an increase in HIV status knowledge for the (study-defined) 'complete package' both in 2018 and 2019, an increase in social support, an increase in generalised self-efficacy and a decrease in the proportion who had at least two lifetime partners. Unexpectedly, in Nairobi there was a suggestion of an increase in condomless sex among sexually active younger AGYW who accessed the complete package compared to sexually active AGYW who were not invited to DREAMS. In Gem, we found evidence of an increase in HIV status knowledge both in 2018 and 2019, and an increase in social support, for the complete package. There was evidence of a reduction in condomless sex and in the proportion with at least two lifetime partners (overall, and among younger AGYW) for the complete package. There was also evidence of a reduction in the proportion with at least one lifetime partner (overall, and among both younger and older AGYW).

For the partial package in Nairobi, we found evidence of an increase in HIV status knowledge both in 2018 and 2019 and an increase in self-efficacy (overall, and among younger AGYW). Unexpectedly, there was an indication of an increase in the proportion with at least one lifetime partner. In Gem, we found evidence of an increase in HIV status knowledge for the partial package in 2018 but not in 2019, an increase in social support (overall, and among younger AGYW) and an increase in self-efficacy (overall, and among younger AGYW). There was evidence of a reduction in condomless sex, and of a reduction in the proportion with at least one lifetime partner (overall, and among both younger and older AGYW).

Taken together, our findings indicate that a package of 5 DREAMS primary interventions in Nairobi, and 4 or 5 in Gem had positive impacts on multiple HIV prevention outcomes. A partial package with fewer interventions (3 or 4 in Nairobi, and 3 in Gem) appeared to have similar effectiveness as the complete package in Gem (for the outcomes that we considered), but was less effective than the complete package in Nairobi. This suggests that the streamlined partial package could offer an efficient alternative to a more comprehensive package in Gem, but a more comprehensive package of interventions is needed in Nairobi informal settlements.

## Interpretation of findings

A recurring question in evaluations of combination HIV prevention interventions relates to what combinations are most effective [22]. From our analysis, we found that a package of 5 DREAMS primary interventions in Nairobi, and 4 or 5 in Gem, had positive impacts on multiple HIV prevention outcomes. This implies that addressing the behavioral, biomedical, and structural drivers of HIV risk among AGYW through an integrated set of interventions is effective and highlights the importance of a comprehensive approach to HIV prevention. The positive impact of both the complete and the partial package may indicate synergistic effects of combining multiple interventions that target behavioural, biomedical and structural aspects, and thereby tackle the multi-dimensional risks faced by AGYW.

We also found that a streamlined package of 3 primary interventions (HIV testing services; school or community-based HIV and violence prevention; and social asset building), addressing each of the 3 broad areas of HIV prevention (biomedical, behavioural, and structural), produced more effects in a rural setting compared to a package of 3 or 4 interventions in an urban setting. This indicates that the effectiveness of the DREAMS package of interventions varies by context. In rural Gem, there is strong community cohesion and more stable household structures, although the area is characterised by limited resources. The similarity of the effectiveness of the streamlined and more complete package in this setting, where resources are limited and few other interventions directed specifically at AGYW exist (previously or concurrently), suggests that an approach that focuses on a specific set of interventions might be suitable. In contrast, urban informal settlements are often characterised by weak social networks due to systemic challenges [23], and AGYW in these settings are frequently engaged in informal work or caregiving responsibilities [14]. Social exclusion, which is common in informal settlements, can foster feelings of helplessness and low self-efficacy, potentially increasing the likelihood of engaging in risky behaviours [24]. Furthermore, these areas are marked by high levels of behavioral risks, primarily arising from inadequate living conditions and severe socio-economic challenges, such as inadequate access to education and employment opportunities [25]. Specifically, the urban informal settlements in Nairobi are characterised by high population density, economic insecurity, household instability, and reduced community connectedness. These factors suggest that a comprehensive package of interventions is necessary to address the unique challenges present in urban informal settlements.

Our findings are consistent with evidence from other evaluations of DREAMS that evaluated impacts of combinations of interventions. For example, in an analysis comparing four intervention components, Mathur and colleagues concluded that two components – educational and asset-building interventions – could have the greatest impact on AGYW's HIV risk [26]. An evaluation study in South Africa found that accessing 3 or more DREAMS interventions contributed to increased HIV testing, attaining higher HIV knowledge index scores and access to contraceptives among AGYW [10]. An analysis comparing AGYW who received 2 or more DREAMS intervention components with AGYW who did not receive the DREAMS

interventions in Lesotho found that access to two or more interventions had an impact on AGYW sexual risk and self-efficacy [11].

Important to note is that effects of the same or similar intervention combinations were not always consistent across outcomes, contexts, or age group. For instance, in Nairobi and Gem we found evidence of an increase in the levels of social support for the complete package among younger AGYW. In Gem, we also found evidence of an increase in social support attributable to the partial package among younger AGYW, whereas in Nairobi this effect was not observed. Among older AGYW, there was evidence of an increase in social support for the complete package in Gem but not in Nairobi, and there was no evidence of an effect for the partial package in either setting. The DREAMS programme used a "layered" approach, with a defined "primary package" of core services tailored to specific age groups to reflect differing developmental needs and risk profiles [16]. These age-based differences in delivery may have contributed to the varying impacts observed across age groups.

Similar to our findings, previous studies on DREAMS have also shown that effects of similar intervention combinations can vary across outcomes and contexts (even within the same study) as well as age group [16,27–31]. This emphasises the widely acknowledged role that context plays in influencing the effectiveness of an intervention, and also the importance of selecting intervention combinations based on the prevalence of the risk factors in that context, the target age group, and the "background" level of service provision.

For both sets of packages analyzed in our study, the focus remained on the individual AGYW. While this targeted "girl-centric" approach has shown positive outcomes, the sustainability of these effects in the long term may rely on wider family, community and institutional support. Family support creates a safe environment, promotes continued education, and facilitates access to services. Community support through the involvement of local leaders and mentors helps shift social norms, thereby enabling AGYW to participate in and benefit from interventions. Institutional support promotes the integration of new services into existing health and education systems. Although the core DREAMS package included family and community level interventions, and interventions for male sexual partners of AGYW, previous evidence suggests that uptake of these interventions remained low [6]. This may explain why impacts on condomless sex were less consistent than the other outcomes, given AGYW cannot change this by themselves. Also, HIV incidence among AGYW in Gem, which is influenced by broader community-wide HIV prevalence and viral suppression – was not much lower during the time frame of this analysis compared with the previous 2–3 years [32].

While some of the observed effects may appear modest, they are practically important. For example, increased knowledge of HIV status is a gateway to timely diagnosis, linkage to care, and prevention of HIV transmission. Improvements in social support and self-efficacy can enhance resilience, increase the uptake of services, and promote behaviour change. Similarly, a reduction in the number of sexual partners lowers the risk of HIV acquisition. These outcomes, though intermediate, are vital for achieving long-term reductions in HIV incidence. These practical gains are essential for programme implementers who are aiming to optimize the real-world impact of DREAMS interventions in diverse settings.

Overall, our findings complement the existing body of research on HIV prevention interventions, extending them to include findings about what combinations of a multi-component intervention can improve outcomes among AGYW. Additionally, our findings support key elements of the DREAMS Theory of Change, which highlighted that a combination of biomedical, behavioral, and structural interventions can synergistically reduce HIV risk and ultimately reduce HIV incidence among AGYW. We observed stronger effects among AGYW who received a more complete package of DREAMS interventions (rather than a partial package) in both settings. This supports the idea that a combination of interventions is essential for shifting drivers of HIV risk and improving outcomes.

## Strengths and limitations

Key strengths of our study include the large representative cohort of AGYW and the high cohort retention rates. Limitations included the potential for residual confounding in our comparisons of non-invitees and those invited who received

the 'partial' or 'complete' package due to unmeasured variables. For example, marital status (which could influence exposure to DREAMS, as well as condom use, number of partners, social support and self-efficacy) was not measured in Gem at cohort enrolment. Despite the high cohort retention rates and controlling for confounding variables measured at enrolment in our analysis, it is also possible that the impact on outcomes differed between individuals who were followed up and those who were not. Another limitation is that we relied on self-reported data and there is a possibility of exposure and outcome misclassification, which could lead to effect sizes being over or under-estimated. For example, the proportion of AGYW in various categories of exposure to DREAMS may be underestimated if AGYW who were invitees did not self-identify as being invited to participate in DREAMS activities or did not accurately report which primary interventions they received. However, exposure misclassification is expected to be small in both settings since the invitation to DREAMS activities was a formalised process that was coordinated by a single implementing partner, and eligibility criteria were clear and well-defined. The process was therefore well understood by both implementers and invitees, making misclassification unlikely. Implementation of DREAMS started in 2016, and there is a possibility that interventions implemented quickly may have impacted some of the confounding variables measured at cohort enrolment (2017/2018), such as school enrolment. However, it is unlikely that this impact was substantial, given the time it took for DREAMS implementation to be fully established [16].

Our measure of lifetime partners may not accurately reflect the intervention's impact if an AGYW had multiple sexual partners before the intervention started. This is particularly important for older AGYW, but less of an issue for younger AGYW (<18 years), the majority of whom were not sexually active at the time of cohort enrolment. Missing data were limited to two confounding variables in Gem, for participants who could not be linked to the external database (HDSS) from which the variables were derived, and such participants were categorised as "unknown" for these two variables (and included in all analyses). The proportion of missingness was relatively low and unlikely to bias the findings.

Although our study was conducted in two Kenyan settings, the findings are likely to be broadly generalisable to similar DREAMS implementation contexts in East Africa. Our results may not be generalisable to all countries where DREAMS is implemented, but because they represent diverse implementation contexts, they can still contribute important insights for other settings in which DREAMS (or other combination HIV prevention) interventions are being delivered. While this study focussed on the impact of receiving either a complete or partial package of DREAMS interventions, future research could aim to understand which interventions contribute most substantially to specific outcomes.

## Conclusion

We found that a "complete" package of interventions that combined behavioural, biomedical, and structural components at the individual level had positive impacts on HIV prevention outcomes among AGYW in two Kenyan settings. Our results emphasise the importance of a comprehensive approach to HIV prevention among AGYW. We also found that a streamlined "partial" package was effective in rural Gem, but not in Nairobi urban informal settlements, highlighting that a one-size-fits-all approach is not appropriate for HIV prevention. Our findings demonstrate the need for context-specific intervention strategies in HIV prevention.

## Supporting information

**S1 Text. Directed Acyclic graphs (DAGS).**
(DOCX)

**S2 Text. Causal inference assumptions and interpretations.**
(DOCX)

**S3 Text. Study procedures for confidentiality, data accuracy, and developmental adaptations.**
(DOCX)

**S1 Table. Combinations of primary interventions by Invitation status in Nairobi and Gem.**
(XLSX)

**S2 Table. Characteristics at enrolment by age group, Nairobi and Gem.**
(XLSX)

**S3 Table. Knowledge of HIV status, 2018 – Sensitivity analyses.**
(XLSX)

**S4 Table. Knowledge of HIV status, 2019 – Sensitivity analyses.**
(XLSX)

**S5 Table. Condomless sex at least once, 2019 – Sensitivity analyses.**
(XLSX)

**S6 Table. Condomless sex, 2019 among Sexually Active AGYW – Sensitivity analyses.**
(XLSX)

**S7 Table. At least one lifetime partner, 2019 – Sensitivity analyses.**
(XLSX)

**S8 Table. Social support, 2019 – Sensitivity analyses.**
(XLSX)

**S9 Table. Generalised self-efficacy, 2019 – Sensitivity analyses.**
(XLSX)

**S10 Table. At least two lifetime partners, 2019 – Sensitivity analyses.**
(XLSX)

## Acknowledgments

We would like to acknowledge the participants in Gem, Korogocho and Viwandani for their time, information, and support. We are also thankful to the project field team and the project and data management staff for their dedication and efforts in ensuring the collection of high-quality data.

## Author contributions

**Conceptualization:** Faith Magut, Sarah Mulwa, Moses Otieno, Elona Toska, Jane Ferguson, Brendan Maughan-Brown, Daniel Kwaro, Abdhalah Ziraba, Isolde Birdthistle, Sian Floyd.

**Data curation:** Faith Magut, Sarah Mulwa, Annabelle Gourlay, Elvis O. A. Wambiya, Moses Otieno.

**Formal analysis:** Faith Magut, Sarah Mulwa.

**Funding acquisition:** Isolde Birdthistle, Sian Floyd.

**Investigation:** Faith Magut, Sarah Mulwa, Brendan Maughan-Brown, Isolde Birdthistle, Sian Floyd.

**Methodology:** Faith Magut, Sarah Mulwa, Isolde Birdthistle, Sian Floyd.

**Project administration:** Vivienne Kamire, Jane Osindo, Daniel Kwaro, Abdhalah Ziraba, Isolde Birdthistle.

**Resources:** Elona Toska, Brendan Maughan-Brown, Daniel Kwaro, Abdhalah Ziraba, Isolde Birdthistle, Sian Floyd.

**Software:** Sian Floyd.

**Supervision:** Vivienne Kamire, Jane Osindo, Daniel Kwaro, Abdhalah Ziraba.

**Validation:** Faith Magut, Sarah Mulwa, Sian Floyd.

**Visualization:** Faith Magut, Sarah Mulwa.

**Writing – original draft:** Faith Magut, Sarah Mulwa.

**Writing – review & editing:** Faith Magut, Sarah Mulwa, Annabelle Gourlay, Vivienne Kamire, Jane Osindo, Elvis O. A. Wambiya, Moses Otieno, Elona Toska, Jane Ferguson, Brendan Maughan-Brown, Daniel Kwaro, Abdhalah Ziraba, Isolde Birdthistle, Sian Floyd.

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
