## [Decision Letter · Decision Letter 0]

9 Jul 2025

PGPH-D-25-00897

What combination of interventions can optimise HIV prevention for adolescent girls and young women? Cohort analysis of DREAMS participation in urban and rural Kenya.

Dear Faith,

Thank you for submitting your manuscript to PLOS Global Public Health. After careful consideration, we feel that it has merit but does not fully meet PLOS Global Public Health’s publication criteria as it currently stands. Therefore, we invite you to submit a revised version of the manuscript that addresses the points raised during the review process.

We look forward to receiving your revised manuscript.

Kind regards,

Collins Otieno Asweto, PhD

Academic Editor

Journal Requirements:

Reviewers' comments:

Reviewer's Responses to Questions

**Comments to the Author**

1. Does this manuscript meet PLOS Global Public Health’s publication criteria ? Is the manuscript technically sound, and do the data support the conclusions? The manuscript must describe methodologically and ethically rigorous research with conclusions that are appropriately drawn based on the data presented.

Reviewer #1: Yes

Reviewer #2: Partly

Reviewer #3: Partly

Reviewer #4: Yes

Reviewer #5: Yes

2. Has the statistical analysis been performed appropriately and rigorously?

Reviewer #1: Yes

Reviewer #2: Yes

Reviewer #3: No

Reviewer #4: Yes

Reviewer #5: Yes

3. Have the authors made all data underlying the findings in their manuscript fully available (please refer to the Data Availability Statement at the start of the manuscript PDF file)?

Reviewer #1: Yes

Reviewer #2: Yes

Reviewer #3: Yes

Reviewer #4: Yes

Reviewer #5: Yes

4. Is the manuscript presented in an intelligible fashion and written in standard English?

Reviewer #1: Yes

Reviewer #2: Yes

Reviewer #3: Yes

Reviewer #4: Yes

Reviewer #5: Yes

5. Review Comments to the Author

Reviewer #1: Overall, this study is excellent and acceptable for publication in PLOS Global Public Health. Several minor comments should be considered to improve the clarity and strength of the article. Please ensure that grammatical errors are addressed throughout the manuscript.

Abstract

- Please describe the DREAMS intervention.

- Should mention the type of statistical approaches used with the causal inference framework (i.e., descriptive statistics, logistic regression).

- There should be a mention of the study design (i.e., a cohort study) in the first sentence of the methods.

- Could you clarify in the abstract that Gem is a rural area and Nairobi is an urban area? The writing in the abstract currently suggests that Nairobi is also a rural area.

- How were people recruited (i.e., online, in-person)? Purposeful sampling/Targeted sampling from settlements?

- Mention that “A total of 1081 AGYW…”, rather than starting a sentence with a number.

- Is self-efficacy in managing their health? This can be clarified.

- Address grammar throughout the abstract, including “This study compares…”.

Introduction

- Need to clarify what “bear the brunt of the epidemic” means. Is it related to HIV infection incidence rates?

- For “inequalities and violence” did you mean to “mitigate/halt inequalities and violence”.

- Briefly connect how DREAMS targets the UNAIDS pillar. Does it touch on all three components?

Methods

- Could you clarify the study design in the first sentence?

- HIV testing services - how would it comprise HIV counselling services? Counselling should be grouped elsewhere.

- Please clarify how social support is defined as a binary variable. This appears to be an aggregate of 4 questions. Is a binary outcome more appropriate?

- A description of Self-efficacy can be more detailed. Is coping associated with mental health, social well-being, and academic/career challenges?

Results

- Aim to avoid starting sentences with numbers.

- Table 2, regarding the p-value for Knowledge of HIV status. This should be for those who access the intervention? Rather than for those who were not invited?

- Ensure that Supplementary tables have previously defined all acronyms used (I.e., HTS).

Discussion

- Based on the literature, are there reasons for increased condomless sex for sexually active individuals accessing the complete package?

- What causes a discrepancy between urban and rural Kenya? Could a more detailed sociocultural context be provided to readers to help them better understand this? For example, what are the “fragmented social challenges” you’ve mentioned in urban areas?

- Clarify “addressing the determinants of HIV risk among AGYW”. This is not clear - what are these determinants? These interventions are not necessarily targeted or tailored to specific risks, but rather constitute a series of broad-spectrum interventions.

- The "findings in context" section appears like a list. These findings could be better integrated throughout the "interpretation of findings" section instead.

- Please change “high levels” to “… high-level behavioural risk”.

- Future steps should also be discussed. The manuscript highlighted the strength of partial interventions, but how will an optimal combination of the Partial DREAMS interventions be identified? Which individual intervention contributed most towards each outcome? This could be conducted in the future. Another consideration should be to offer a tailored set of DREAMS interventions for each person. Should tailored interventions be selected based on some form of risk-stratification model?

Reviewer #2: The manuscript potentially makes a contribution to the HIV prevention literature by examining combination intervention packages rather than individual interventions. The focus on DREAMS is relevant considering the substantial financial investment into this initiative. Also, using data from established demographic surveillance systems provides a good base for the analysis.

The outcomes are generally well-defined and measured using appropriate timeframes.

Concerns requiring revision

1. Self-reported intervention receipt has been solely used without validation against program records or other objective measures. Self-reporting is subject to recall bias, bias due to perceived social desirability, or misunderstanding of the components of the intervention. Although acknowledged in the limitations, an assessment of concurrence between self-reported and program-records of intervention receipt would have been useful in avoiding any potential for misclassifying exposure status.

• If possible, validation of self-reported intervention receipt vis-à-vis program records for a subset of participants. It will also be important to acknowledge the limitations of self-reported exposure measurement and discuss the potential direction and magnitude of bias from miscalculation of exposure.

2. The manuscript does not discuss missing data patterns, mechanisms, or handling strategies. There was no reporting of missing data percentages for key variables. Was missing data missing at random (MAR) or missing not at random (MNAR). What was the impact of missing data assumptions?

• Provide detailed missing data analysis including percentages missing for all key variables and assess patterns of missingness. Discuss potential impact of missing data on study conclusions.

3. The exposure variable ‘invited by 2018 and accessed 0, 1, or 2 primary interventions’ suggests that accessing 0 intervention in equivalent to accessing 2. In the context of combination prevention, 2 interventions can be potentially significant in influencing outcomes depending on what and for whom.

• It would be helpful to reflect on how it may or not have impacted results and the question of the value of investing in two interventions if you can achieve same result with having only been invited to DREAMS.

4. There is insufficient detail on the timing of the intervention receipt relative to measurement of the outcome. Did all participants have the outcomes measured before completing interventions? Considering that the duration of intervention exposure varied across participants, there is a potential for reverse causation.

• Provide a timeline to show intervention implementation periods and outcome measurement windows. Analyses should be restricted to participants with adequate follow-up time after intervention completion. Discuss potential for reverse causation and strategies used to address it.

5. Numerous statistical tests have been conducted across multiple outcomes and subgroups without adjustment for multiple comparisons. Five primary outcomes tested in two settings with age stratification results in over 20 statistical tests. There may be an increased risk of Type I error (false positive findings)

• Apply appropriate multiple testing corrections, distinguish between primary and secondary analyses and acknowledge the increased risk of Type I error in the interpretation.

6. The generalizability of findings beyond the study settings is not adequately discussed. Both settings are in Kenya.

• Expand discussion of to include factors that may influence generalizability. For example, comparison of the study settings to other implementation sites based on other characteristics beyond urban/rural. It would also be useful to reflect on the generalizability to other countries.

7. Strengthen connection between the findings and the DREAMS theory of change as well as more detailed discussion of implementation factors that may explain the differences.

8. More detailed discussion on the practical importance of observed effects, beyond statistical significance, is needed. Consider including discussion on the practical significance of observed effects for program implementation. Also, contextualizing the findings in relation to other evaluations of DREAMS.

9. Given the changing financial landscape, cost-effectiveness implications would be very useful.

Recommendation

The findings from this study have the potential to inform DREAMS programming and broader HIV prevention efforts for AGYW, but the limitations identified above should be addressed to ensure the conclusions are robust and reliable.

Reviewer #3: I commend the authors for a good job in presenting the DREAMS study analysis. My comments from the submission are as follows:

(1) What methods were used to prevent "contamination" of interventions to those who were not invited to participate in the study; or was this taken care of in the statistical analysis.

(2) The authors mentioned using DAG to draw causal assumptions between exposure and outcomes, the DAG was not presented in the paper. Can this DAG be presented in the paper?

(3) Authors used 2 methods for regression, multivariable regression and propensity score regression. Is there any particular reason for using these at the same time? If one method is better than the other, then it makes sense to use that method. If both are similar then presenting one method still makes sense?

(4) When using propensity score regression for causal analysis there are several assumptions that are made. Can the authors state and show if these assumptions were met, and if not what methods were used to assess for the violation of assumptions. For example SUTVA (stable unit treatment value assumption) and exchangeability.

(5) Tables can be improved so that the results presented are better to read and understand.

Reviewer #4: The manuscript is well constructed, cohesive and delineates the principal interventions targeting AGYW.

Please provide additional details on the following:

1) Please clarify the mean (or median) age difference between questionnaire administrators and adolescent girls and young women (AGYW) who provided informed consent.

2) Describe the physical setting in which the questionnaires were administered e.g., a private room versus a communal household space, and indicate whether parents, caregivers, or other AGYW were within visual or auditory range during enrolment and data collection.

3) Detail the procedures used to safeguard confidentiality when sensitive items were addressed.

4) Outline the methods employed to verify response accuracy and specify any quality-control metrics applied.

5) Explain the analytical or design strategies adopted to adjust for confounding variables such as participant age, mental-health status, cultural practices, and religious affiliation during risk profiling and stratification.

6) Given the broad eligibility age range of 10–24 years, how were questions related to lifetime number of sexual partners structured to ensure age-appropriateness and cognitive suitability for younger participants? Please describe any adaptations made to account for developmental differences across early, middle, and late adolescence.

Reviewer #5: Overall comments

The authors provide a comprehensive evaluation of a combination of interventions that can optimize HIV prevention among adolescent girls and young women (AGYW) in Kenya. This work contributed to the evidence base on HIV programming in resource-limited settings. However, this manuscript could be made stronger and could benefit from streamlining of the content and improved clarity.

See comments below for consideration

Introduction

Clearly outlined with excellent contextual information. However, it’s a bit long and would benefit from a reduction in some content that can be moved to the methods section. For example, the DREAMS selection process and package lists could be moved to the methods section.

Methods

Well-structured design applied across two diverse settings. Detailed explanation and account of the methods provided. While the use of statistical techniques is appropriate and required, the analysis section is overly lengthy and somewhat technical, which may limit some audiences. Perhaps consider summarizing the key analytic steps more concisely to improve clarity and reader engagement.

Ethical considerations-where were the consent forms stored?

Discussion

The findings are appropriately synthesized and interpreted within the broader literature.

Line 517-555 provides a summary of the findings, though long. Please revise to be more concise and focus on the key findings.

Line 551-552- Why do you think the partial package was effective than the complete package in Nairobi?

Line 586-593 Briefly discuss how the intervention tailoring by age can influence these patterns

Line 594-596: Provide details on how the sustainability of the effects in the long term may rely on family, community, and institutional support. Integration?

6. PLOS authors have the option to publish the peer review history of their article (what does this mean? ). If published, this will include your full peer review and any attached files.

**Do you want your identity to be public for this peer review?** For information about this choice, including consent withdrawal, please see our Privacy Policy .

Reviewer #1: No

Reviewer #2: No

Reviewer #3: No

Reviewer #4: No

Reviewer #5: No

---

## [Decision Letter · Decision Letter 1]

17 Sep 2025

What combination of interventions can optimise HIV prevention for adolescent girls and young women? Cohort analysis of DREAMS participation in urban and rural Kenya.

PGPH-D-25-00897R1

Dear Faith,

We are pleased to inform you that your manuscript 'What combination of interventions can optimise HIV prevention for adolescent girls and young women? Cohort analysis of DREAMS participation in urban and rural Kenya.' has been provisionally accepted for publication in PLOS Global Public Health.

Best regards,

Collins Otieno Asweto, PhD

Academic Editor

Reviewer #1:

Reviewer #4:

Reviewer #5:

Reviewer Comments (if any, and for reference):

Reviewer's Responses to Questions

**Comments to the Author**

1. If the authors have adequately addressed your comments raised in a previous round of review and you feel that this manuscript is now acceptable for publication, you may indicate that here to bypass the “Comments to the Author” section, enter your conflict of interest statement in the “Confidential to Editor” section, and submit your "Accept" recommendation.

Reviewer #1: All comments have been addressed

Reviewer #4: All comments have been addressed

Reviewer #5: All comments have been addressed

2. Does this manuscript meet PLOS Global Public Health’s publication criteria ? Is the manuscript technically sound, and do the data support the conclusions? The manuscript must describe methodologically and ethically rigorous research with conclusions that are appropriately drawn based on the data presented.

Reviewer #1: Yes

Reviewer #4: Yes

Reviewer #5: Yes

3. Has the statistical analysis been performed appropriately and rigorously?

Reviewer #1: Yes

Reviewer #4: Yes

Reviewer #5: Yes

4. Have the authors made all data underlying the findings in their manuscript fully available (please refer to the Data Availability Statement at the start of the manuscript PDF file)?

Reviewer #1: Yes

Reviewer #4: Yes

Reviewer #5: Yes

5. Is the manuscript presented in an intelligible fashion and written in standard English?

Reviewer #1: Yes

Reviewer #4: Yes

Reviewer #5: Yes

6. Review Comments to the Author

Reviewer #1: Thank you for the opportunity to read and review this excellent article. One final comment, there were a couple extra spaces and periods in the text. Please ensure that the grammar is addressed prior to publication.

Reviewer #4: (No Response)

Reviewer #5: No more comments for the authors. My key concerns were addressed

7. PLOS authors have the option to publish the peer review history of their article (what does this mean? ). If published, this will include your full peer review and any attached files.

**Do you want your identity to be public for this peer review?** For information about this choice, including consent withdrawal, please see our Privacy Policy .

Reviewer #1: No

Reviewer #4: No

Reviewer #5: No
